# PEGA-BA@Ce6@PFCE Micelles as Oxygen Nanoshuttles for Tumor Hypoxia Relief and Enhanced Photodynamic Therapy

**DOI:** 10.3390/molecules28186697

**Published:** 2023-09-19

**Authors:** Junan Zhang, Xiaoyun Jiang, Wenyue Luo, Yongjie Mo, Chunyan Dai, Linhua Zhu

**Affiliations:** 1College of Chemistry and Chemical Engineering, Hainan Normal University, Haikou 571158, China; 20212070300024@hainnu.edu.cn (J.Z.); 202007070724@hainnu.edu.cn (W.L.); 202212070300015@hainnu.edu.cn (Y.M.);; 2Key Laboratory of Tropical Medicinal Resource Chemistry of Ministry of Education, Haikou 571158, China; 3Key Laboratory of Functional Organic Polymers of Haikou, Haikou 571158, China

**Keywords:** hypoxia, block copolymer, oxygen delivery, photodynamic therapy

## Abstract

Tumor hypoxia, which is mainly caused by the inefficient microvascular systems induced by rapid tumor growth, is a common characteristic of most solid tumors and has been found to hinder treatment outcomes for many types of cancer therapeutics. In this study, an amphiphilic block copolymer, poly (ethylene glycol) methyl ether acrylate-block-n-butyl acrylate (PEGA-BA), was prepared via the ATRP method and self-assembled into core-shell micelles as nano radiosensitizers. These micelles encapsulated a photosensitizer, Chlorin e6 (Ce6), and demonstrated well-defined morphology, a uniform size distribution, and high oxygen loading capacity. Cell experiments showed that PEGA-BA@Ce6@PFCE micelles could effectively enter cells. Further in vitro anticancer studies demonstrated that the PEGA-BA@Ce6@PFCE micelles significantly suppressed the tumor cell survival rate when exposed to a laser.

## 1. Introduction

Hypoxia represents a commonly observed phenomenon in tumor cells [1,2]. This occurrence stems from an intricate interplay between inadequate oxygen supply and aberrant oxygen consumption within the tumor microenvironment. Hypoxia leads to decreased oxygen tension in tumor tissues [3,4], the abnormal proliferation of tumor cells [5,6], and angiogenic dysfunction [7,8]. Severe tumor hypoxia can result in decreased efficacy of tumor treatment [9,10], and even metastasis and drug resistance [11,12,13,14]. Therefore, hypoxia has become a significant challenge in cancer treatment. Currently, there are two primary ways to improve tumor hypoxia in cells in vitro using nanoparticles. One way is to carry exogenous oxygen. For instance, Tang [15] developed a hybrid semiconducting organosilica-based O_2_ nanoeconomizer, pHPFON-NO/O_2_, to combat tumor hypoxia. Cheng [16] developed a versatile nanoplatform (HAFOE-Ce_6_/Pt-NPs(O_2_)) with rapid and controlled release of its loaded oxygen. Zhuang [17] synthesized a chitosan (CS)-capped nanosystem that could codeliver oxygen, photosensitizer, and chemotherapy drugs. The other method to improve tumor hypoxia is endogenous oxygen production. Wang [18] reported a metal–organic framework, Mn_3_[C(CN)_6_]_2_, that serves as a catalase-like nanozyme for oxygen generation. Chen [19] reported a multifunctional therapeutic platform, GOx@MBSA-PPy-MnO_2_, NPs that catalyzes the transformation of H_2_O_2_ into hydroxyl radicals (·OH) and O_2_ via a Fenton-like reaction, effectively relieving tumor hypoxia.

Photodynamic therapy (PDT) has gained increasing interest as a cancer treatment strategy. In PDT, a photosensitizer (PS) absorbs energy from lasers and converts oxygen to cytotoxic reactive oxygen species (ROS), directly killing tumor cells [20,21]. However, the efficacy of PDT is largely affected by hypoxia in solid tumors. At the same time, PDT exacerbates hypoxia by consuming oxygen, preventing it from fully realizing its potential [22]. Block polymers have been widely developed as artificial oxygen carriers for enhancing PDT. Block polymers are known for their excellent self-assembly behaviors in aqueous solutions, allowing for the encapsulation of both hydrophobic and hydrophilic molecules in tailored nanostructures [23,24]. Therefore, block polymers have been utilized to enhance PDT by carrying oxygen in photosensitizer-containing nanoparticles. Several amphiphilic fluoropolymers have been developed into nanocapsules or nanomicelles and loaded or conjugated with photosensitizers like chlorin e6 (Ce6), IR780, and boron-dipyrromethene (BODIPY) to improve cancer PDT [25,26,27,28]. 

In this study, a unique type of amphiphilic block polymer was fabricated to enable tumor hypoxia relief and efficient tumor PDT (Figure 1). In our design, polymers were synthesized by the ATRP using EBIB as a macro-initiator between two organic building blocks, PEGA and BA. Nuclear magnetic resonance (NMR) and gel permeation chromatograph (GPC) were used to detect the yield and molecular weight of PEGA-BA. Dynamic light scattering (DLS) and transmission electron microscopy (TEM) were used to observe the particle size and morphology of PEGA-BA. Owing to the excellent self-assembly capability of block polymers, perfluoro-15-crown-5-ether (PFCE) and photosensitizer Chlorin e6 (Ce6) can be loaded into the hydrophobic core of PEGA-BA. Ce6 can efficiently generate toxic 1O2 under 660 nm laser irradiation, while PFCE will serve as a reliable oxygen carrier to alleviate the hypoxic tumor microenvironment because of its high oxygen-carrying capability. Afterward, we synthetically investigated the oxygen-carrying capacity, drug-loaded and released capacity, and ROS generation capacity of the fabricated micelles. The results from our vitro experiments suggest that our synthesized micelles effectively enhanced the PDT effect.

## 2. Results and Discussion

### 2.1. Composition of PEGA-BA

PEGA-BA was synthesized through a two-step ATRP using PEGA and BA. The structure of PEGA-BA was identified using ^1^H NMR (Figure 1A) and ^13^C NMR (Figure 1B). The ^1^H-NMR spectra of PEGA-BA exhibited signals at 0.75–1.00 (k) ppm, which represented the methyl groups of BA [29]. The proton signals of CH-CH_2_ appeared at 1.75–2.10 (g) and 2.10–2.50 (f) ppm [30], indicating that the PEGA-BA block polymer was successfully synthesized. Based on ^13^C NMR spectroscopy, the carbon signal of CH-CH_2_ was observed at 40–42 (f) and 63–65 (d) ppm, indicating the occurrence of polymerization reactions.

The compositions of the prepared samples were analyzed using FT-IR spectra. Figure 1C shows the FTIR patterns of BA, PEGA, PEGA_10_-BA_40_, and PEGA_15_-BA_20_. In comparison with the typical peaks of PEGA and BA, the spectrum of PEGA-BA exhibited a disappearing absorption peak at 1201 and 987 cm^−1^, which indicated the polymerization of the carbon–carbon double bond. The characteristic peaks of PEGA at 1349 cm^−1^ were also observed. Additionally, the characteristic peaks of BA at 1276 cm^−1^ were observed [31], confirming the successful linkage of PEGA with BA.

Furthermore, the molecular weight of PEGA-BA was identified using GPC (Figure 2A), and the conversion of polymerization values were 85% and 90%, resulting in PEGA-BA with Mn (SEC) values of 8646 and 11528, and dispersity values of 1.089 and 1.149, respectively.

### 2.2. Physicochemical Characterizations of PEGA-BA Micelles

The particle size of PEGA-BA micelles was determined via DLS and ranged from 10 to 100 nm, as shown in Figure 1B,C. The small size of the particles aided in the polymer passing through the tumor tissue and delivering oxygen to the tumor. The stability of PEGA-BA in an aqueous solution was confirmed through seven days of testing (Figure 2D), with the results indicating that the micelles were particularly stable in an aqueous solution.

The morphology of PEGA-BA micelles was further investigated via TEM, and the images revealed a uniform spherical structure with particle sizes distributed around 50 nm (Figure 2E,F).

**Figure 2 molecules-28-06697-f002:**
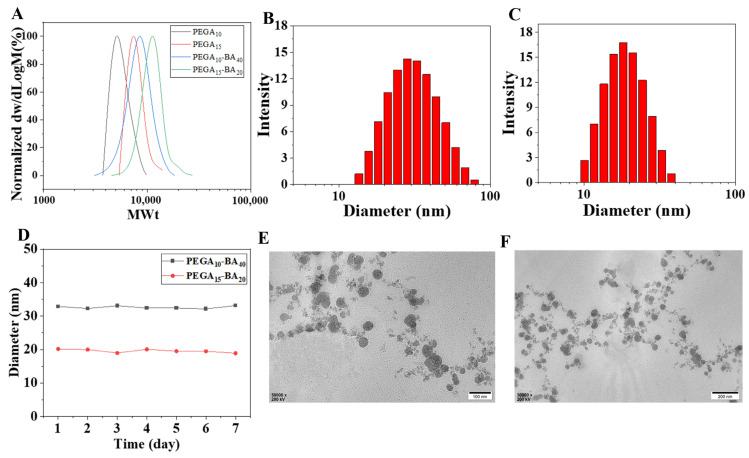
Particle size of PEGA-BA micelles. (**A**) The molecular weight of PEGA_10_-BA_40_ and PEGA_15_-BA_20_. DLS result of (**B**) PEGA_10_-BA_40_ and (**C**) PEGA_15_-BA_20_. (**D**) Size change of PEGA-BA in PBS according to DLS analysis. (**E**) TEM image of PEGA_10_-BA_40_ and (**F**) PEGA_15_-BA_20_. Results exhibit that the morphology of PEGA_15_-BA_20_ was a sphere. Scale bar, (**E**) 100 nm and (**F**) 200 nm.

### 2.3. Drug Loading and Release

The UV-vis absorption spectra were used to measure the drug loading and release of PEGA-BA. Due to the difference in water solubility between DOX and Ce6 and the limited loading capacity of PEGA-BA, excess DOX and Ce6 were sonicated to load them into the hydrophobic core of PEGA-BA. The solution was dialyzed in physiological saline for 48 h, and the absorbance was determined via UV-vis to plot the standard curve. To determine the concentrations of DOX and Ce6, samples were taken at different time periods and the change in solution concentration was measured using a UV-visible spectrophotometer (Figure 3A,B). It can be seen from the figure that after 48 h of dialysis, the loading efficiency of PEGA_10_-BA_40_ was much higher than that of PEGA_15_-BA_20_. Figure 3C,D show a comparison between the samples after dialysis for 48h and a solution of known concentration. Values of 400 nm and 500 nm were, respectively, taken as the absorption peaks of Ce6/DOX, and a standard curve was drawn to calculate the loading capacity of PEGA_10_-BA_40_ and PEGA_15_-BA_20_ (Figure 3E,F). As shown in Figure 3E,F, the content of Ce6 and DOX in the PEGA_10_-BA_40_ micelles after dialysis was 9.7 μg/mL and 4.2 μg/mL; meanwhile, the loading capacities of PEGA_15_-BA_20_ were only 4.9 μg/mL and 2.4 μg/mL. Then, we simulated the acidic microenvironment in tumors to place the dialyzed solution in pH 5 physiological saline, and tested the drug release ability of PEGA-BA. The absorbance changes in the micelles were measured via UV-vis spectroscopy at different time intervals. The drug release curves of PEGA-BA@Ce6@DOX were plotted (Figure 3G,H). As can be clearly seen from the figure, there was a significant difference in the release rate of PEGA_10_-BA_40_@Ce6@DOX and PEGA_15_-BA_20_@Ce6@DOX in acidic and neutral solutions. After 48 h, the drug release rate of PEGA_10_-BA_40_@Ce6@DOX reached 60%, which was higher than that of PEGA_15_-BA_20_@Ce6@DOX 40%. Furthermore, it was observed that the capacity of PEGA-BA to load DOX was notably lower than Ce6, leading to the abandonment of DOX in our subsequent experiments.

### 2.4. Measurement of O_2_ Release

Van der Waals interactions between oxygen and fluoride enable the dissolution of a substantial amount of oxygen by fluoride. To investigate the oxygen-carrying capacity and release behavior, the change in oxygen concentration was monitored after adding PEGA-BA micelles and admitting oxygen for 10 min. The results showed that the addition of PEGA_10_-BA_40_@PFCE and PEGA_15_-BA_20_@PFCE with oxygen loading resulted in a significant increase in the dissolved oxygen concentration in the related systems (Figure 4A). However, solutions with the addition of H_2_O and PBS pretreated with nitrogen bubbling showed a rapid descent in their dissolved oxygen concentrations. The oxygen content in H_2_O decreased to 14 mg/mL, while the oxygen content in PEGA_15_-BA_20_ was maintained at 18 mg/mL. These results indicate that PEGA_10_-BA_40_@PFCE and PEGA_15_-BA_20_@PFCE are efficient in oxygen loading.

### 2.5. Measurement of ROS Generation

PDT promotes the production of ROS, leading to increased cellular damage [32,33]. To assess this, the ^1^O_2_ and cellular ROS generation capabilities were tested using SOSG and DCFH-DA as probes, respectively. SOSG was used to primarily monitor the content of ^1^O_2_, and fluorescence intensity increased when SOSG was oxidized by ^1^O_2_. As shown in the figure (Figure 4B), after the irradiation of PEGA-BA@Ce6@PFCE micelles for 10 min, there was a 1.5-fold enhancement in fluorescence intensity as a result of increased oxygen dissolution achieved by PEGA-BA, as compared to free Ce6. Meanwhile, fluorescence intensity in H_2_O was barely detected. This suggests that PEGA-BA efficiently delivers oxygen to enhance PDT efficacy.

To further measure the intracellular ROS production of MCF-7, CLSM was used with DCFH-DA as the ROS index. When oxidized by ROS to DCF, the probe emitted bright green fluorescence. Specifically, the oxygen-loaded micelles were first added to a glass substrate and incubated for two hours. After cell absorption, the ROS probe DCFH-DA was added. Following 20 min of incubation, the cells were subjected to 5 min of laser irradiation to induce ROS production, after which they were examined using CLSM. Figure 4C,D display CLSM images at varying magnifications, revealing that cells treated with PEGA10-BA40@Ce6@PFCE and PEGA15-BA20@Ce6@PFCE exhibited intense green fluorescence upon laser irradiation. In contrast, no significant fluorescence intensity was detected in the PBS group. This demonstrates that PEGA-BA@Ce6@PFCE micelles can efficiently generate ROS and enhance PDT efficacy.

### 2.6. In Vitro Photo-Toxicity Efficacy

The ROS generated can directly damage cell membranes and induce tumor cell death [34,35]. Therefore, an MTT assay was conducted to verify the cytotoxicity of different formulations. To rule out the possibility that enhanced cytotoxicity of PEGA_10_-BA_40_ and PEGA_15_-BA_20_ formulations was introduced by the copolymer itself, cell viability was also evaluated (Figure 5A). Further analysis of the cytotoxicity of PEGA10-BA40 and PEGA15-BA20 formulations loaded with Ce6 revealed that PEGA10-BA40@Ce6 and PEGA15-BA20@Ce6 exhibited lower toxicity than free Ce6 (Figure 5B). This observation suggests that PEGA10-BA40@Ce6 and PEGA15-BA20@Ce6 achieved a sustained release of Ce6. Blank and control groups were set up to verify the photodynamic effect. Significant cytotoxicity was observed in MCF-7 cells treated with PEGA10-BA40@Ce6@PFCE and PEGA15-BA20@Ce6@PFCE following 10 min of continuous laser irradiation (Figure 5D). Conversely, no evident cytotoxicity was observed in the blank group (Figure 5C). The concentration of micelles was further diluted to investigate the minimum toxicity of the micelles on cells (Figure 5E). The results indicated that a concentration of micelles greater than 19 μg/mL had a significant inhibitory effect on the cells. The reason for the different PDT effects may be that larger micelles have larger hydrophobic cores, which can encapsulate more drugs and oxygen, thus enhancing the PDT effect.

## 3. Materials and Methods

### 3.1. Materials

Poly(ethylene glycol) methyl ether acrylate (PEGA), Ethyl 2-bromoisobutyrate (EBIB), and Tris [2-(dimethylamino) ethyl] amine were purchased from Sigma-Aldrich (Shanghai, China). *n*-Butyl acrylate and Isopropyl alcohol (IPA), 3-(4,5-Dimethylthiazol-2-yl)-2,5-diphenyltetrazolium bromide (MTT), Chlorin e6(Ce6), and Perfluoro-15-crown-5-ether (PFCE) were obtained from Macklin (Shanghai, China). Fetal bovine serum (FBS) was purchased from Every Green (Hangzhou, China). Dulbecco’s Modified Eagle Medium (DMEM), Phosphate-Buffered Saline (PBS), and Trypsin were purchased from Solarbio (Beijing, China). 2′,7′-Dichlorodihydrofluorescein diacetate, Singlet Oxygen Sensor Green, and Hoechst 33342 were obtained from Beyotime (Shanghai, China). All of the chemicals were used as supplied without further purification.

### 3.2. Cell

MCF-7 human breast cancer cells were cultured at 37 °C under a humidified 5% CO_2_ atmosphere and supplied with DMEM, which was supplemented with 10% FBS, 100 U/mL of penicillin, and 100 mg/mL of streptomycin.

### 3.3. Synthesis of PEGA-BA

In the polymerization process of PEGA-BA, Ethyl 2-bromoisobutyrate (EBIB, 0.150 g, 1.13%) served as the initiator. The hydrophilic block, Poly (ethylene glycol) methyl ether acrylate (PEGA, 1.844 g, 14.00%); the solvent, isopropyl alcohol (IPA, 5.246 g, 39.85%); and the catalyst, Tris[2-(dimethylamino)ethyl]amine (0.021 g, 0.16%) were added into a sample bottle equipped with a magnetic stirring bar. The bottle was sealed with a rubber plug, purged with nitrogen for 15 min, and subjected to stirring at 600 rpm for a duration of 6 h. Subsequently, the hydrophobic monomer (BA, 5.9 g, 44.82%) was incorporated, and the reaction was maintained at room temperature for 18 h [36,37]. The final solution was dialyzed (MWCO = 3500 Da) for 72 h against IPA and unreacted products removed. After purification, the product was lyophilized and stored at normal atmospheric temperature.

### 3.4. Preparation of PEGA-BA@Ce6@PFCE Micelles

To prepare the drug-loaded micelles, 5 mg of Ce6, 5 mg of DOX and 50 μL of PFCE, and 20 mg of PEGA-BA were accurately weighed and co-dissolved in 100 μL of DMSO. The solution was added dropwise to 4 mL of distilled water with ultrasonication for 15 min. The micelles were then dialyzed in a dialysis bag (MWCO = 3500 Da) against distilled water for 48 h to remove organic solvents and unencapsulated drugs. Subsequently, 50 μL of PFCE was added to PEGA-BA via ultrasonication. Finally, the dialyzed drug-loaded micelles were collected in a centrifuge tube and stored at 4 °C for later use.

### 3.5. Physicochemical Characterization of Drug Loaded PEGA-BA Micelles

The chemical compositions of PEGA-BA dissolved in CDCl_3_ were analyzed using ^1^H NMR and ^13^C NMR (Bruker AV-400 MHz, Zurich, Switzerland) operating at 400 MHz. Samples of 10 mg were dissolved in 0.6 mL of CDCl_3_ and homogenized via ultrasound. The FT-IR spectra of various samples were recorded using a Nicolet AVATAR 360 FTIR spectrometer in the frequency range of 400–4000 cm^−1^.

The molecular weight was determined via gel permeation chromatography (GPC, Shimadzu LC-20A, Kyoto, Japan). Briefly, 30 mg samples were dissolved in 1 mL THF and homogenized via ultrasound. The samples were placed in a 1.5 mL sample bottle and again homogenized via ultrasound.

The particle size of the micelles was analyzed via dynamic light scattering (DLS, Malvern Zetasizer Nano ZS90, Birmingham, UK). A 10 mg/mL solution of PEGA-BA was prepared, and 1 mL of the solution was placed into a transparent cup after being dissolved evenly via ultrasound. The specific morphology of the micelles was observed using transmission electron microscopy (TEM, JEOL JEM-2100plus, Tokyo, Japan). PEGA-BA was formulated into a 1 mg/mL solution, which was evenly dissolved via ultrasound, and then, dropped onto an organic copper web for examination.

### 3.6. Drug Release of PEGA-BA Micelles

Considering the acidic microenvironment commonly found in tumors, we dialed drug-loaded micelles in a pH 5 physiological saline solution to simulate the drug release of micelles in the tumor. Specifically, 5 mL 200 μg/mL of drug-loaded micelles was added to 4 L of acidic physiological saline for dialysis. Samples were taken regularly to measure the concentration changes using UV-vis spectroscopy and determine the drug release rate of micelles in a pH 5 acidic environment.

### 3.7. Measurement of O_2_ Release

The oxygen content in the aqueous solution was measured using a portable dissolved oxygen meter (YSI 550A, Yellow Springs, OH, USA) [38]. To determine the oxygen-carrying ability of PEGA-BA, 15 mL of distilled water was first flushed with nitrogen for 10 min to remove any pre-dissolved oxygen. Then, 45 mg of PEGA-BA was added to the solution, and oxygen was introduced by flushing it with oxygen gas for 15 min. An oxygen probe was then placed into the solution, and the oxygen concentration was recorded using the dissolved oxygen meter for 20 min.

### 3.8. In Vitro ROS Generation

To evaluate the singlet oxygen production capability of PEGA-BA@Ce6@PFCE micelles at the solution level, 5 mg/mL of PEGA-BA@Ce6 micelles was combined with 25 μL of PFCE and mixed with 15 μM SOSG. After laser irradiation for 10 min, the fluorescence intensity at 525 nm was measured using a fluorometer (HITACHI F-7100,Tokyo, Japan). To perform intracellular ROS release detection, a DCFH-DA probe was used. MCF-7 cells were seeded in laser confocal dishes at a density of 1 × 10^5^ cells per well. After overnight incubation, 1 mL of the formulation solution (each dish containing 4 μg/mL of Ce6) was added to the cells. The cells were then incubated in a 5% CO_2_ incubator at 37 °C. After 2.5 h, 1 mL of DCFH-DA probe was added and incubated with the cells for 30 mins. The cells were then washed twice with PBS and irradiated with a 660 nm laser for 1 min per dish. After washing the cells with PBS, they were stained with Hoechst 33342 (0.01 mM) at 37 °C for 10 min and washed with PBS three times. The cell images were captured using a confocal laser scanning microscope (Nikon A1+T, Tokyo, Japan).

### 3.9. In Vitro Photo-Toxicity Efficacy

To measure the survival rate of MCF-7 cells after treatment with different formulations, an MTT assay was conducted with different concentrations of free Ce6, PEGA-BA, and PEGA-BA@Ce6@PFCE. The cells were incubated in a 5% CO_2_ incubator or an anoxic incubator overnight before the experiment, and both laser and non-laser irradiation groups were set up. After 2 h of incubation, the cells were exposed to a laser for 10 min. The MCF-7 cells were cultured for 24 h, and the phototoxicity of the various materials was investigated using the MTT assay. In brief, 10 μL of 5 mg/mL MTT solution was added to the treated cells, and the cells were incubated for 4 h in a 5% CO_2_ incubator at 37 °C. The supernatant was aspirated and 150 μL of DMSO was added to each well. Then, the absorbance at 570 nm was measured using a Multifunctional enzyme marker (Bio Tek Synergy HTX, Winooski, VT, USA), and the cell survival rate was calculated. Moreover, the toxicity of the material was also examined. Different concentrations of PEGA-BA were prepared, and the MCF-7 cells were incubated for 24 h after the addition of the PEGA-BA solution. Finally, the cell viability was examined using the MTT assay.

### 3.10. Statistical Analysis

All data are expressed as mean ± standard error of the mean. The statistical difference between different data groups was evaluated using one-way ANOVA, and a *p*-value of less than 0.05 was considered to be statistically significant. Asterisks (*) are used to denote statistical significance between bars (* *p* < 0.05, ** *p* < 0.01, *** *p* < 0.001). The statistical analysis was performed using GraphPad Prism 9.1.

## 4. Conclusions

In summary, this study delineates a novel amphiphilic block copolymer with high oxygen-loading capacity for enhancing PDT effectiveness. The PEGA-BA micelles exhibit a uniform size, favorable morphology, and remarkable stability. Upon encapsulating Ce6 and PFCE, they effectively relieve hypoxic conditions within tumors. Additionally, the PEGA-BA@Ce6@PFCE micelles generate ROS under 660 nm laser irradiation, facilitating tumor cell death. This research underscores the potential of the PEGA-BA block copolymer, which offers advantages such as straightforward synthesis and robust oxygen-carrying capabilities. These attributes collectively relieve tumor hypoxia and amplify PDT efficacy. This study provides a valuable reference for the application of block copolymers in cancer therapy.

## Data Availability

Data will be made available on request.

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
