# Peer review of "PEGA-BA@Ce6@PFCE Micelles as Oxygen Nanoshuttles for Tumor Hypoxia Relief and Enhanced Photodynamic Therapy"

_molecules, 2023, doi:10.3390/molecules28186697_

Round 1

Reviewer 1 Report

Comments and Suggestions for Authors

The communication submitted by Zjang et al. deals with the synthesis of two amphiphilic copolymer samples and the study of their self-assembly. Moreover, a photosensitizer was loaded and different in vitro tests were carried out.

The manuscript is clear, well written and the conclusions are supported by the results. However, some corrections are needed in order to increase the quality before publication:

1. the authors must use the term “micelles” instead of “nanoparticles” as it is more specific for their system.

2. the introduction section can be completed with some references concerning the drug-loaded micellar systems.

3. the authors must move the results concerning the GPC from 3.2 to section 3.1. the formation of block copolymer can be assessed only by GPC. Moreover, the spectra of the precursor PEGA should be added in order to demonstrate the Mn increase.

4. in fig 3 E, why there is such a big difference between the 2 samples at concentration higher than 19 μg/ml?

5. which are the loading and encapsulation efficiencies?

6. drug release rate can be also evaluated.

Comments on the Quality of English Language

English is ok.

Author Response

Dear reviewer,

Thank you for your  comments concerning our manuscript entitled “PEGA-BA@Ce6@PFCE Micelles as Oxygen Nanoshuttles for Tumor Hypoxia Relief and Enhanced Photodynamic Therapy”. We have carefully studied your comments and have made correction which we hope meet with your approval. And some improvements and clarifications of our paper,please refer to the document.

Sincerely

Chunyan Dai

Reviewer 2 Report

Comments and Suggestions for Authors

Author Response

Dear reviewer,

Thank you for your  comments concerning our manuscript entitled “PEGA-BA@Ce6@PFCE Micelles as Oxygen Nanoshuttles for Tumor Hypoxia Relief and Enhanced Photodynamic Therapy”. We have carefully studied your comments and have made correction which we hope meet with your approval.And some improvements and clarifications of our paper,please refer to the document.

Sincerely

Chunyan Dai

Round 2

Reviewer 1 Report

Comments and Suggestions for Authors

The authors have taken into consideration almost all the comments. However, at the end of the introduction section they must indicate the analysis techniques which were used for the characterization of their system. In fig 1, replace "self-assemble" with "self-assembly".

Author Response

Dear reviewer,

Thank you for your comments concerning our manuscript entitled “PEGA-BA@Ce6@PFCE Micelles as Oxygen Nanoshuttles for Tumor Hypoxia Relief and Enhanced Photodynamic Therapy”. We have carefully studied your comments and have made correction which we hope meet with your approval. I would be very grateful if this manuscript could be considered for publication after this revision.

Yours sincerely

Chunyan Dai

Reviewer 2 Report

Comments and Suggestions for Authors

The x-axis in Fig 3E and F is signed still "concertation" and it should be "concentration". Please correct

Comments on the Quality of English Language

The text still requires minor English language mistakes

Author Response

(The authors gave the same response as above.)
